# Metformin as a Treatment Strategy for Sjögren’s Syndrome

**DOI:** 10.3390/ijms22137231

**Published:** 2021-07-05

**Authors:** Joa Kim, Yun-Sung Kim, Sung-Hwan Park

**Affiliations:** 1Division of Rheumatology, Department of Internal Medicine, Chosun University Hospital, Gwangju 61453, Korea; sjoan16@naver.com (J.K.); daniel117@chosun.ac.kr (Y.-S.K.); 2Division of Rheumatology, Department of Internal Medicine, College of Medicine, The Catholic University of Korea, Seoul 06591, Korea

**Keywords:** Sjögren’s syndrome, metformin, AMPK/mTOR pathway

## Abstract

Sjögren’s syndrome (SS), a chronic inflammatory disease involving the salivary and lacrimal glands, presents symptoms of sicca as well as systemic manifestations such as fatigue and musculoskeletal pain. Only a few treatments have been successful in management of SS; thus treatment of the disease is challenging. Metformin is the first-line agent for type 2 diabetes and has anti-inflammatory potential. Its immunomodulatory capacity is exerted via activation of 5’ adenosine monophosphate-activated protein kinase (AMPK). Metformin inhibits mitochondrial respiratory chain complex I which leads to change in adenosine mono-phosphate (AMP) to adenosine tri-phosphate (ATP) ratio. This results in AMPK activation and causes inhibition of mammalian target of rapamycin (mTOR). mTOR plays an important role in T cell differentiation and mTOR deficient T cells differentiate into regulatory T cells. In this manner, metformin enhances immunoregulatory response in an individual. mTOR is responsible for B cell proliferation and germinal center (GC) differentiation. Thus, reduction of B cell differentiation into antibody-producing plasma cells occurs via downregulation of mTOR. Due to the lack of suggested treatment for SS, metformin has been considered as a treatment strategy and is expected to ameliorate salivary gland function.

## 1. Introduction

Sjögren’s syndrome (SS) is an autoimmune disease involving salivary and lacrimal glands. Although one to 10 per 1000 individuals have been reported to suffer from the disease, no effective treatment has been suggested [1]. Both innate and adaptive immune response have been considered responsible for the pathogenesis of SS [2]. Environmental triggers, such as viral infection, upregulate interferon alpha (IFN-α) in the mucosal epithelial cells of individuals with susceptible genetic factors for SS. This increase in IFN-α causes enhanced secretion of B cell activating factors (BAFF), resulting in autoantibody production within ectopic germinal center (GC)-like structures. In addition, epithelial cells produce autoantigens and form immune complex (IC), leading to production of more IFN-α [3]. In consequence, this immune response results in damage to tissue of the target organs.

Management of SS is challenging and no specific guidelines have been suggested due to the heterogenetic manifestation of the disease. Metformin (dimethyl biguanide), a representative hypoglycemic agent used worldwide for type 2 diabetes, has anti-inflammatory and immunomodulatory capacities. 5’ adenosine monophosphate-activated protein kinase (AMPK) has been suggested as a key regulator of this anti-inflammatory effect [4]. AMPK regulates the inflammatory pathway via inhibition of nuclear factor-κB (NF-κB) and downregulation of the Janus kinase/signal transducer and activator of transcription (JAK/STAT) signaling pathways [5].

The aim of this review is to evaluate the pathogenesis of SS and to enumerate anti-inflammatory and immunomodulatory properties of metformin, an old drug with significant potential. Furthermore, we present a new aspect of the drug and suggest it as a strategy for treatment of SS.

## 2. Metformin and Its Anti-Inflammatory and Immunomodulatory Effect

Metformin, a representative drug for treatment of diabetes, is used worldwide. It controls hyperglycemia by reducing hepatic gluconeogenesis via AMPK-dependent and AMPK-independent mechanisms [6,7,8,9,10]. AMPK is an important energy sensor in maintenance of cellular energy homeostasis. The mitochondrial respiratory chain complex, which is bound in the inner layer of mitochondrial membrane, catalyzes nicotinamide adenine dinucleotide (NADH) oxidation which leads to synthesis of adenosine tri-phosphate (ATP). This series of processes is known as oxidative phosphorylation [11]. Metformin inhibits mitochondrial respiratory chain complex Idirectly and, as a result, causes depletion of ATP and an increase of adenosine mono-phosphate (AMP) [10,12]. This change in AMP to ATP ratio promotes activation of AMPK [6] which leads to ATP-generating catabolic pathways and simultaneously inhibits ATP-consuming pathways [13].

The mammalian target of rapamycin (mTOR), a downstream molecule of AMPK signaling, is an important regulator of T cell differentiation. mTOR-activated T cells differentiate into effector T cells, such as T helper 1 (Th1), T helper 2 (Th2), and T helper 17 (Th17) cells while mTOR deficient T cells differentiate into regulatory T cells (Treg) [14,15]. Delgoffe et al. [15] demonstrated that metformin specifically targets mTOR complex 1 (mTORC1), which was proven to influence Th1 and Th17 cells. Interestingly, metformin was found to improve obesity-induced inflammation. Brown adipose tissue (BAT) is known to delay diet-induced obesity and inhibit development of metabolic disorders. Fibroblast growth factor 21 (FGF21), a main regulator for obesity, reduces obesity-mediated inflammation. FGF21 was recently reported to have reduced inflammatory cytokine release and diminished the Th17- interleukin (IL)-17 axis through the STAT3 pathway [16,17]. Metformin increased FGF21 levels in BAT and thus ameliorated inflammation in obese collagen-induced arthritis (CIA) mice [18].

Although T cells are mainly known to contribute in many autoimmune diseases, B cells play a crucial role in many autoimmune conditions. In one study, conditional deletion of the mTOR gene in B cells significantly diminished B cell proliferation and GC differentiation [19]. As a result of IL-21, CD154 and other stimuli, signal transducer and activator of transcription 3 (STAT3) is activated, resulting in differentiation of B lymphocytes into plasma cells [20,21]. Metformin suppresses STAT3 activation through the AMPK/mTOR pathway and, as mentioned, STAT3 activation results in differentiation of B cells into antibody-producing plasma cells. Using *Roquin*
^san/san^ mice as a murine model for systemic lupus erythematosus (SLE), Lee et al. verified that metformin may improve kidney and liver inflammation in addition to reduced anti-ds DNA antibody production [22].

Macrophages may polarize in two phenotypes, “classically activated” M1 and “alternatively activated” M2. M1 macrophages mainly produce pro-inflammatory cytokines such as tumor necrosis factor-alpha (TNF-α), IL-1, IL-6, IL-12, IL-23 and monocyte chemoattractant protein-1 (MCP-1) as M2 macrophages produce anti-inflammatory cytokines such as IL-10 and transforming growth factor beta (TGF-β) [23]. Metformin was associated with altering macrophage polarization to anti-inflammatory M2 macrophages via the AMPK pathway in addition to reduction of IL-6 and TNF-α levels [24]. Increased expression of mTOR is a signature phenomenon found in gout patients. Autophagy of monosodium urate (MSU) crystals in macrophages leads to release of IL-1ß, resulting in activation of NLRP3-inflammasome [25,26,27]. Moreover, MSU crystals enhance the mTOR pathway, release IL-1ß, IL-6, IL-8, and IL-18 as well as provoke cell death in monocytes in vitro. Therefore, mTOR inhibitor metformin may inhibit the inflammatory process in gout patients [28].

In the state of stress conditions such as inflammation, activation of nicotinamide adenine dinucleotide phosphate (NADPH) leads to generation of reactive oxygen species (ROS) and, consequently, neutrophil extracellular traps (NETs) are released. NETs consist of DNA, histones, and antimicrobial peptides, which are regarded as a major source for SLE pathogenesis. NETosis refers to a specific form of cell death in neutrophils and formation of NETs [5,29]. Knight et al. demonstrated this widely accepted theory with a New Zealand mixed 2328 (NZM) mouse model of SLE [30]. Metformin reduces NADPH oxidase and downregulates the NET pathway. In addition, an in vitro study showed that metformin diminished release of NET DNA and reduced generation of CpG-stimulated plasmacytoid dendritic cells- (PDCs) induced IFNα [31].

Due to their immunomodulatory and anti-inflammatory capacities, mesenchymal stem cells (MSCs) are considered as a treatment for autoimmune disease [32]. The therapeutic effect of adipose-derived mesenchymal stem cells (Ad-MSCs) toward autoimmune disease was equivalent to that of bone marrow mesenchymal stem cells (BM-MSCs) [33]. Ad-MSCs cause cell-to-cell contact and secretion of soluble factors such as hepatocyte growth factor, prostanglandin-E2 (PGE2), indoleamine 2, 3-dioxygenase (IDO), IL-10, and TGF-β, leading to suppression of pro-inflammatory T cell proliferation and regulation of immune response [34,35]. Interferon gamma (IFN-γ) promotes overexpression of STAT1 and causes suppression of MSC-mediated T cells [36]. Vigo et al. demonstrated that mTOR inhibition leads to phosphorylation of STAT1 and STAT3, which mediates IFN-γ- induced immunoregulation [37]. In this manner, Jang et al. showed that STAT1 phosphorylation by metformin promotes the immunoregulatory effect of MSCs in Murphy Roths Large-lymphoproliferation strain (MRL/*lpr*) mice which develop severe SLE manifestation [38]. In addition, another study demonstrated the anti-inflammatory and chondroprotective capacities of metformin-stimulated Ad-MSCs in osteoarthritis (OA) mouse models [39].

Metformin appears to be effective in treatment of ankylosing spondylitis (AS), an autoimmune disease; fibroblasts play a critical role in the pathogenesis [5]. TNF-α is the key cytokine in the pathogenesis of AS, and TNF inhibitor is the mainstream of its treatment. Metformin blocks TNF-α through the AMPK pathway. Results of a recent study showed that administration of metformin resulted in reduced osteogenic-specific markers and inhibited ossification. Metformin inhibits ossification of fibroblasts in many ways. First, it blocks release of IL-6, which is thought to suppress fibroblast ossification. Moreover, AMPK also alleviates ossification and, as mentioned above, metformin enhances AMPK [40].

Due to these anti-inflammatory and immunomodulatory effects, metformin has been proven effective in animal models of autoimmune diseases and inflammatory diseases such as SLE, rheumatoid arthritis (RA), OA, gout, and SS, in addition to the treatment of diabetes, and recently, human clinical trials have been attempted (Table 1).

## 3. Immune Response Concerning Gut Flora Modulation May Be Effective on Sjögren’s Syndrome

Gut microbiota refers to a microorganism found in the gastrointestinal tract. It is mainly found in distal intestine and consists of more than 1000 species [63]. Many factors attribute to the composition of gut microbiota. In neonates, mode of delivery or feeding appears to have an effect, while in adults, dietary behavior and body mass index (BMI) may result in certain strains inhabiting the gut [64]. It is noteworthy that individuals vary in microbiota composition since certain strains may be related to obesity or diabetes [65,66]. Certain organisms have increased glucose intolerance; for example, reduction of *Lactobacilli* may cause glucagon-like peptide I (GLP-1) resistance [67].

Certain gut microbiota are considered to have an association with specific cytokines, which enhances the inflammatory cascade [64]. Therefore, individuals with autoimmune diseases, such as spondyloathritis, RA, Behçet’s disease, and SS, were found to have altered gut flora compared to healthy controls [68,69]. In particular, in SS, *Pseudobutyrivibrio*, *Escherichia/Shigella*, and *Streptococcus* were increased and *Bacteroides*, *Parabacteroides*, *Faecalibacterium,* and *Prevotella* were relatively reduced. Furthermore, reduced diversity of the gut microbiota was observed in patients with severe SS [70].

As confirmed in many studies, metformin caused alterations in the intestinal microbiota [71,72]. A study showed an expanded lifespan in *Caenorhabditis elegans* due to altered folate metabolism in intestinal bacteria by metformin [73]. The concentration of metformin in the gut and in the serum is different. It is 100-fold higher in the gut lumen, which might imply that metformin concentration may be affected by gut microbiota [74].

Changes in the intestinal microbiota are well known in SS, and the therapeutic effect of administration of the bacterial metabolite butyrate has been proven in an animal model of SS [74]. Therefore, further research on metformin and its immunomodulatory effect via change in gut flora and its metabolites is necessary.

## 4. Pathophysiology of Sjögren’s Syndrome

Th17 cells, which are important for inflammation response at mucosal barriers, and their representative cytokine IL-17, are found in salivary glands of SS patients. Th17 cells and T-follicular helper (Tfh) cells play a critical role in primary SS especially at disease onset when balance between these effector T cells and their regulatory counterparts (i.e., Treg cells and T-follicular regulatory (Tfr) cells) is deranged [75].

Furthermore, it seems that the STAT3 pathway participates in pathogenesis of SS. Vartoukian et al. showed that overexpression of pro-inflammatory cytokine IL-17 in SS patients was due to reduction of suppressor of cytokine signaling 3 (SOCS3), which is a negative regulator for the JAK/STAT3/IL-17 pathway [76].

As demonstrated by Shah et al., rapamycin, the most potent mTOR inhibitor, improved autoimmune dacryoadenitis in an SS animal model [77]. In addition, rapamycin caused a decrease in B cell proliferation and IgG production in SS patients [78]. Finally, metformin via STAT3 inhibition suppressed Tfh and B cell differentiation and diminished GC response [22].

GC-like structures are found in target organs of SS patients. They produce autoantibodies such as anti-Ro/SSA and anti-La/SSB, resulting in loss of glandular function and increased risk of lymphoma [79,80]. Tfh cells and B cells are important in GC structures [81]. STAT3-deficient CD4^+^T cells had a defect in Tfh cell differentiation, which resulted in reduction of GC B cells and production of antigen-specific antibodies [82].

Metformin inhibited differentiation of CD4^+^ T cells into Th17, Th1, and Tfh cells, and conversely promoted them into their regulatory counterparts, such as Treg and Tfr cells. Metformin reduced GC B cell and serum IgG levels and induced balance between IL-10 and IL-17 producing B cells. These effects of metformin are mediated via the AMPK/mTOR/STAT3 pathway [22,43,44,83].

To this mechanism, metformin is thought to be a strategy for treatment of SS (Figure 1).

## 5. Treatment of Sjögren’s Syndrome

In management of SS, clinicians may find that treatment of the disease is difficult because the symptoms are hard to control and there is no cure. SS-related comorbidities, such as fibromyalgia, anxiety, and depression can influence the severity of the disease, and this may lead to the need for a multidisciplinary approach in management of SS [27,84]. Only a few well-designed, controlled studies concerning treatment for SS have been reported.

For management of oral manifestations, topical fluoride use, gustatory and masticatory stimulation and chlorhexidine administration are recommended. Dry eye treatment is suggested according to cause and severity. Disease-modifying anti-rheumatic drugs (DMARDs) have been suggested for extraglandular manifestations of SS [85].

Targeted therapy (e.g., B cell, pro-inflammatory cytokine, Janus kinase) based on pathogenesis has recently been attempted for the treatment of SS, which does not respond to these conventional DMARDs. Rituximab has been reported to be relatively effective when used in patients with acute parotid gland swelling, not responding to corticosteroid, polysynovitis, pulmonary involvement, peripheral neuropathy, and cutaneous vasculitis [85].

There is an urgency to find other strategies for management of SS. Due to the fact that imbalance between Th17, Tfh cells and Treg, Tfr cells participate in the pathogenesis of SS, metformin has been suggested as a therapeutic option. Metformin was proven to suppress autoimmune dacryoadenitis [77]. Kim et al. showed that salivary flow rate significantly increased in metformin-treated mice at week 20, suggesting that metformin improves salivary gland function when administered for a certain amount of time. Metformin enhances Treg cells and downregulates Th1 and Th17 cells via the AMPK/mTOR pathway. In addition, metformin had anti-inflammatory and immunomodulatory effects without causing hypoglycemia in SS mouse models [62].

## 6. Conclusions

The pathophysiology of SS is still unclear and ambiguous, and no specific treatment guidelines have been suggested. Many attempts have been made to understand the disease, but the results were unsatisfactory. Furthermore, recent randomized trials of rituximab, B-cell depleting agent, have failed. As a result, there is a necessity for new treatment.

Metformin, an anti-hyperglycemic agent, has been used for decades and its safety as a drug has been verified. Other aspects of the drug have recently been observed. Its immunoregulatory effects gave rise to the possibility for its use in autoimmune diseases.

Metformin has been considered as a treatment for many autoimmune diseases such as SLE, RA, and even gout. However, these attempts have been limited to experimental studies and well-designed randomized trials are necessary. For the need of novel treatment for SS, metformin may serve as a promising therapeutic strategy.

## Figures and Tables

**Figure 1 ijms-22-07231-f001:**
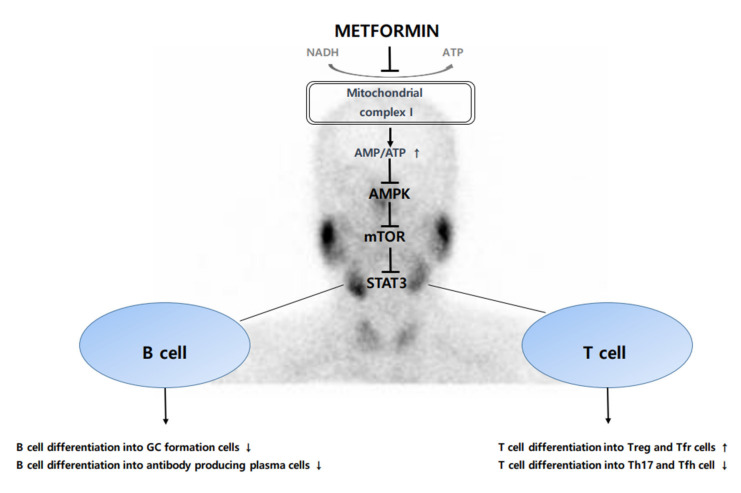
Schematic diagram of anti-inflammatory and immunomodulatory effects of metformin.

**Table 1 ijms-22-07231-t001:** Experimental animal and human studies concerning metformin and its anti-inflammatory effect in autoimmune and inflammatory disease.

Systemic lupus erythematosus	AMPK/mTOR/STAT3 regulation leads to suppression of B cell differentiation into plasma cells in Roquin ^san/san^ mice [22]. Administration of metformin enhanced AMPK, STAT1 expression, and downregulated STAT3, mTOR in Ad-MSCs and, as a result, improved disease activity in MRL/lpr mice [38]. Proof-of-Concept Trial of metformin established an association between mtDNA in NETs, anti-mtDNA antibodies, and PDC IFNα pathogenesis in SLE [31]. Normalization of CD4^+^ T cell metabolism via glycolysis and mitochondrial metabolism inhibition improved lupus activation in vitro and in vivo [41]. Post hoc analyses from two randomized trials revealed that metformin reduced disease flares in patients with SLE [42].
Rheumatoid arthritis	Metformin had an anti-inflammatory effect via inhibition of Th17 cell differentiation in a CAIA murine model [43]. Diminished IL-17 producing Th17 cells, enhanced Treg cells, suppressed osteoclastogenesis in a CIA model [44]. Metformin and CoQ10 combination therapy showed more improved joint inflammation by reduction of Th17 cells, induction of Treg cells and inhibition of osteoclastogenesis than metformin or CoQ10 alone in a CIA murine model [45]. Enhanced AMPK, FGF21 production and BAT differentiation resulted in reciprocal Th17/Treg balance and improved CIA in a murine model of diet-induced obesity [18]. IGF-IR/PI3K/AKT/mTOR pathway caused inhibition of RA-FLS proliferation, which is important in development of RA [46]. Rapamycin-metformin reduced clinical arthritis score and ameliorated the metabolic profile in obese mice with CIA [47]. Insulin resistance is linked to both BMI and synovitis in RA and metformin reduced GLUT-1 expression in synovial tissue from RA patients [48]. Treatment with metformin lowered the risk of RA in a retrospective cohort study [49]. Combination therapy with metformin and COX-2 inhibitor lowered the admission rate of T2DM patients with RA [50]. The inhibitory effects of metformin (e.g., suppressed osteoclastogenesis and reduced expression of inflammatory cytokines) on RA pathogenesis were investigated in vitro [51]. Metformin inhibited degradation of the cartilage-layer matrix, osteoclast formation, and chondrocyte apoptosis [52]. In combination with LMT-28, which suppressed IL-6 mediated signaling, metformin improved arthritic score in CIA mice [53]. Metformin proved to be beneficial when combined with MTX in a double-blind placebo-controlled study of RA patients.AMPK activator (metformin) promoted an increase in HAPLN1 level in RA-FLS [54].
Osteoarthritis	Upregulated autophagy resulting from enhanced SIRT1 protein expression by metformin alleviated cartilage degradation in an OA mouse model [55]. Metformin promoted activation of AMPK, resulting in NF- κ B inhibition in IL-1β-induced ATDC5 cells and protected chondrocytes [56]. Articular cartilage change in type 2 diabetes can be protected by metformin via an anti-inflammatory effect in a mouse model [57]. Administration of metformin after DMM surgery restricted OA development and progression via AMPK activation [58]. Obese OA patients had reduced disease progression in a prospective cohort study [59]. Combination therapy of metformin and COX2 inhibitor reduced the joint replacement surgery rate in OA and type 2 diabetic patients [60]. Metformin ameliorated structural worsening and pain in mouse models [61]. Metformin-stimulated Ad-hMSCs had a greater antinociceptive and chondroprotective effect than unstimulated Ad-hMSCs [39].
Gout	Metformin inhibited mTOR signaling leading to a reduction in cell death and a decrease in inflammatory mediators from MSU crystal- stimulated monocytes [28].
Sjögren’s syndrome	Metformin improved salivary gland function by regulating T cells and B cells in a mouse model [62].
Ankylosing spondylitis	Suppressed ossification and inflammation in AS fibroblasts [40].

AMPK, 5′-adenosine monophosphate-activated protein kinase; mTOR, mammalian target of rapamycin; STAT3, signal transducer and activator of transcription 3; Ad-MSC, adipose-derived mesenchymal stem cell; Mt DNA, mitochondrial DNA; NETs, neutrophil extracellular traps; PDC, plasmacytoid dendritic cells; IFNα, interferon alpha; SLE, systemic lupus erythematosus; CD4, cluster of differentiation 4; Th 17 cell, T helper 17 cell; CAIA, collagen antibody-induced arthritis; IL-17, interleukin-17; Treg cell, regulator T cell; CIA, collagen-induced arthritis; CoQ10, coenzyme Q 10; FGF21, fibroblast growth factor 21; BAT, brown adipose tissue; IGF-IR, insulin-like growth factor 1 receptor; PI3K, phosphoinositide kinase 3; RA-FLS, rheumatoid arthritis fibroblasts-like synoviocytes; BMI, body mass index; GLUT-1, glucose transporter 1; COX-2, cyclooxygenase-2; T2DM, type 2 diabetes mellitus; IL-6, interleukin-6; MTX, methotrexate; HAPLN1, hyaluronan and proteoglycan link protein 1; SIRT1, sirtuin 1; OA, osteroarthritis; NF-κB, nuclear factor kappa-light-chain-enhancer of activated B cells; IL-1β, interleukin 1 beta; DMM, destabilization of the medial meniscus; Ad-hMSC, adipose tissue-derived human mesenchymal stem cell; MSU, monosodium urate.

## Data Availability

Not applicable.

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
