# Peer review of "Metformin as a Treatment Strategy for Sjögren’s Syndrome"

_ijms, 2021, doi:10.3390/ijms22137231_

Round 1

Reviewer 1 Report

The current review is an interesting contribution to the field of treatment of Sjogren's syndrome. Despite this, some concerns need to be addressed:

Plagiarism check using Turnitin revealed a level of 30%. This is relatively too high. The authors need to re-work their manuscript so the levels will decrease below 20% (the allowed threshold).  

This paper lacks a figure showing and explaining Metformin's mechanism via 5' adenosine 13 monophosphate-activated protein kinase (AMPK) activation. The authors have described it in the text; however, it is not informative. Doing one good figure is of interest to all the readers of IJMS.

Many sentences throughout the review lack adequate reference(s) lines 163-165, 171-176, and many more.. The authors need to add a reference for every sentence.

Reviewer 2 Report

This submitted review is very informative, discussing in great detail the potential therapeutic effects of metformin for Sjögren’s Syndrome from the view point of its characteristics and the function of metformin. The flow of the argument is not particularly problematic. However, Figure 1, which is listed in LINE 212, is not found in this manuscript. In addition, there are many cases where the verb change according to the subject (e.g., LINE 31 cause -> causes) is not done correctly. In addition, some grammatically unfavorable expressions are found, such as LINE 46 'Furthermore, presenting a new aspect of the drug and suggest it as a treatment strategy for SS'. These points need to be added or corrected.

Round 2

Reviewer 1 Report

the reviewers answered my comments